# Examining the Impact of Artificial Intelligence and Social and Computer Anxiety in E-Learning Settings: Students' Perceptions at the University Level

Mohammed Amin Almaiah [1,2,3,*], Raghad Alfaisal [4], Said A. Salloum [5], Fahima Hajjej [6], Sarah Thabit [7], Fuad Ali El-Qirem [8], Abdalwali Lutfi [9], Mahmaod Alrawad [9], Ahmed Al Mulhem [10], Tayseer Alkhdour [1], Ali Bani Awad [10] and Rana Saeed Al-Maroof [7]

[1] Department of Computer Networks, College of Computer Sciences and Information Technology, King Faisal University, Al-Ahsa 31982, Saudi Arabia
[2] Faculty of Information Technology, Applied Science Private University, Amman 11931, Jordan
[3] King Abdullah the II IT School, Department of Computer Science, The University of Jordan, Amman 11942, Jordan
[4] Faculty of Art, Computing and Creative Industries, Universiti Pendidikan Sultan Idris, Tanjong Malim 35900, Malaysia
[5] School of Science, Engineering, and Environment, University of Salford, Manchester M50 2EQ, UK
[6] Department of Information Systems, College of Computer and Information Sciences, Princess Nourah bint Abdulrahman University, P.O. Box 8442, Riyadh 11671, Saudi Arabia
[7] English Language and Linguistics Department, Al Buraimi University College, Al Buraimi 512, Oman
[8] Faculty of Architecture and Design, Al-Zaytoonah University of Jordan, Amman 11733, Jordan
[9] College of Business, King Faisal University, Al-Ahsa 31982, Saudi Arabia
[10] College of Education, King Faisal University, Al-Ahsa 31982, Saudi Arabia
* Correspondence: malmaiah@kfu.edu.sa or m_almaiah@asu.edu.jo

**Abstract:** The learning environment usually raises various types of anxiety based on the student's abilities to use technology and their abilities to overcome the negative feelings of an individual being watched all the time and criticized. Hence, learners still feel anxious while using computers and socializing in an e-learning environment. Learners who are faced with computer and AI tools are confused and frustrated. The uneasiness stems from anxiety or uneasiness, which is highly evident in daily interaction with computers and artificial intelligence tools or devices in e-learning contexts. The uneasiness stems from anxiety or uneasiness, which is highly evident in the daily interaction with computers and artificial intelligence tools or devices in e-learning contexts. To investigate this phenomenon empirically, a questionnaire was distributed among a group of undergraduate students who are studying different majors. This study aims to investigate the role of social anxiety and computer anxiety in an e-learning environment at the university level. Universities in the Gulf area are among those implementing e-learning systems. In spite of this, recent studies have shown that most students at Gulf universities are still resistant to using online systems; hence, it is necessary to determine the type of anxiety that creates such resistance and their relationship with other external variables such as motivation, satisfaction and self-efficacy. Students would be more likely to use e-learning tools and participate more effectively in their courses using the accessible electronic channels when the degree of anxiety is low. In this study, we have proposed a theoretical framework to investigate the role of social anxiety and computer anxiety in e-learning environments in the Gulf region. We examined how different variables such as satisfaction, motivation and self-efficacy can negatively or positively affect these two types of anxiety.

**Keywords:** social and computer anxiety; self-efficacy; motivation; satisfaction

## 1. Introduction

The use of technology may lead to unpleasant side effects, which may include strong, negative emotional feelings that arise due to the uneasiness of the situation and the novelty

of the tools in educational contexts, especially when learners are interacting with computers at the early stage of learning [1,2]. The anxiety that is developed due to the usage of different tools in the educational environment may lead to frustration and confusion, and these consequences may appear during daily class interaction causing social anxiety. Social anxiety arises from the daily interaction and communication with both learners and teachers in an e-learning environment [3–6]. Learning styles in an e-learning environment seem to be affected by the two concepts of anxiety and motivation. Studies have shown that positive effects may lead to a commitment and motivation to learning, whereas negative effects may result in anxiety, which may finally lead to low achievement and low results [7,8]; therefore, motivation and anxiety are two distinguishing factors that affect learning styles.

Hence, when dealing with artificial intelligence and social and computer anxiety, attention should be paid to the three types of anxiety, namely, trait, state and concept-specific. Though these types have critical differences, they may contribute to computer anxiety from different perspectives [9]. Trait anxiety is related to the apprehension that is developed towards life experience. State anxiety, on the other hand, arises due to the learner's learning background. Concept-specific is related to a specific situation. Therefore, computer anxiety is a concept-specific anxiety that implies that learners' anxiety appears due to the specific context where computers are used [10,11]. Previous studies have provided a direct relationship between computer anxiety and computer use. They proposed that a high level of computer anxiety may lead to computer-use difficulty. However, learners that lack computer anxiety may develop better skills which affect their academic achievement positively [8,12–15]. Social anxiety and computer anxiety are considered dependent variables which are affected by a group of external variables affecting the level of anxiety towards the e-learning environment. One of these factors is computer self-efficacy, which appears when learners are introduced to technology during earlier stages of learning. Thus, an e-learning environment may decrease or increase students' anxiety. Based on the literature, there is strong evidence that anxiety may affect academic achievements. Therefore, students with a high level of anxiety tend to have low academic performance and vice versa [16–19]. Despite the fact that most foreign language learning literature focuses on the achievement of learners, only a considerable number of previous studies concentrate on learners' perceptions of the e-learning environment and anxiety [20,21]. How an e-learning environment can diminish their anxiety due to social interaction and anxiety in using computers and other innovative tools has not been investigated. When two types of anxieties are in cooperation, the process of learning in an e-learning environment will be accounted for more logically and empirically, suggesting that learning can occur when there is an effective social environment and less computer literacy. In other words, a gap is found in the prior research in examining the impacts of external factors on three dependent variables: artificial intelligence and social and computer anxiety. The significance of this study lies in its aims to explore the effect of anxiety on learners' tendency to learn effectively in an e-learning environment, which may affect language learning objectives. The necessity for trustworthy environments for EFL language learning and teaching forms significantly influences the extensive use of these technologies in language instruction [22,23].

Based on the limited utility of existing prior research on artificial intelligence anxiety (AIA) and its relationship with the other two types of anxiety, the objective of this research is to examine the perceptions of the learners towards these types of anxiety. To be able to evaluate the nature and scope of artificial intelligence and social and computer anxiety, several different dimensions must be defined in terms of the proposed framework. The crucial components are as follows: (1) Artificial intelligence anxiety is integrated with immersion, interaction and imagination to form a clear conception of learners' attitudes; (2) social anxiety aspects are measured by integrating it with motivation and satisfaction; (3) computer anxiety is related to self-efficacy; and (4) the inclusion of these different aspects in relation to perceived ubiquity and innovativeness to measure the intention to use technology.

## 2. Literature Review

*Artificial Intelligence and Social and Computer Anxiety*

Artificial intelligence anxiety (AIA) has been the concern of a few studies recently. AIA is utilized to evaluate learners' perception of AI technology, which shows the overall distribution of the levels of learners' AIA levels as rated by other people. AI technology is shortly expected to grow remarkably. Learners' Perceptions of AIA technology may have a negative impact on the success of AI development. Hence, minimizing the level of perceived anxiety may promote the expanded use of AI technology to alter future users' perceptions [24]. Another study by [25] focused on the difference between computer anxiety and artificial intelligence anxiety, making a sharp distinction between the rich environment that is represented by AI technology in contrast to computer anxiety that has limited set of forms in its hardware and software. It has concluded that various factors can affect AI anxiety and proposes eight factors, namely, privacy violation anxiety, bias behaviour anxiety, job replacement anxiety, learning anxiety, existential risk anxiety, ethics violation anxiety, artificial consciousness anxiety and lack of transparency anxiety. In addition to the fact that there are four sources of AI anxiety, uncertainty artificial consciousness anxiety and lack of transparency anxiety also exist.

Studies have shown that AI is different from computers because AI technology is based on autonomous decisions existing in various virtual shapes whereas computers are controlled by humans lacking a rich variety of forms. Hence, computers exist as hardware and software and they do not have personalized service characteristics, which stand in contrast with AI [25,26]. With reference to computer anxiety, studies have proposed that computer anxiety plays an important role in developing learners' abilities and is closely related to self-efficacy. The lack of good computer skills may negatively affect the learners' achievement, leading to a higher level of anxiety [8,14,27]. There are studies that have tackled computer anxiety in relation to the Technology Acceptance Model (TAM) model, which was originally developed by [28], who set a model that contains the basic variables that can measure technology acceptance or adoption, whereas other studies adopted the Theory of Reasoned Action (TRA) model. Both of them can provide detailed results on the importance of computer anxiety from learners' perceptions. They have concluded that computer anxiety is affected by various factors such as self-efficacy, enjoyment, internet experience, perceived ease of use and perceived usefulness. The e-learning environment is significantly controlled by computer anxiety, which has a close relationship with attitude and experience. Learners with positive attitudes and high computer experience will show a low level of computer anxiety. On the other hand, learners who adopt negative attitudes towards computer skills with low experience tend to have a higher level of computer anxiety [8,14,27].

Similarly, previous studies on social anxiety have been conducted to measure the effect of social anxiety on learners' engagement, achievement and performance. They have shown that there is a significant effect of social anxiety on interaction in an e-learning environment. Social anxiety (SASE) has been used to tackle computer anxiety studies, whereas the hypothesis conceptual model was used for other studies. Both methods could provide detailed results for the impact of social anxiety on the e-learning environment. SASE refers to the "Social Anxiety Scale for E-Learning Environments". It is a scale developed by [3] to measure social anxiety and its impact on interaction with the e-learning environment. The results have proven that the lack of confidence in learners and the fear of being misunderstood by others results in a high level of social anxiety. Reducing social anxiety and fear will result in increasing the interaction in an e-learning environment. Additionally, increasing their interaction with e-learning will result in developing their outcomes of e-learning [3,29]. In other words, students can be less anxious and stressed in an e-learning environment when they feel confident and are not worried about being misjudged by others. Table 1 illustrates the previous studies that have focused on computer anxiety and social anxiety.

**Table 1.** Studies on Artificial Intelligence and Social and Computer Anxiety.

| Study Name/Author | Type of Anxiety | Models | The Aim | The Sample | The Outcome |
|---|---|---|---|---|---|
| [24] | Artificial Intelligence Anxiety | N/A | To explore the scale of AIA | University Students | The increasing importance of artificial intelligence necessitates the need to reduce the anxiety that appears as a result of using AI technologies |
| [25] | Artificial Intelligence Anxiety | N/A | The study aims to explain the source of AI anxiety. | A total of 494 valid samples of male and female respondents | The study explained the source of anxiety defining eight AI factors and classifying them into four dimensional pathways. |
| [14] | Computer anxiety | Acceptance model | The purpose of this study is to examine the interaction between computer anxiety and e-learning self-efficacy, in part through the interaction between computer anxiety and e-learning self-efficacy. | University students' | To moderate the effect of anxiety on perceived ease of use, computer self-efficacy is an important factor. |
| [13] | Computer anxiety | Analyzing related literature | Research factors that cause computer anxiety determine how to reduce it by identifying effective treatment options. | Literature review | 1. Learning effectiveness can be affected by computer anxiety. 2. In order to create positive e-learning experiences, human resources need to pay more attention to this anxiety and adopt the appropriate treatments. It will be beneficial to a lot of computer users if computer anxiety can be effectively reduced. |
| [30] | English learning anxiety | Multimedia technology | This study seeks to assess the effect of e-learning teaching in the classroom. | EFL university students | Students can be less anxious and stressed in a multimedia classroom environment. English teachers can use multimedia tools to help students in improving their English proficiency and reduce their language anxiety. |
| [31] | Social-evaluative anxiety | Hypothesis conceptual model | In their study, researchers attempted to determine whether there was a relationship among students' learning flows and the outcomes of their learning during the pandemic in South Korea. | Nursing students | To improve nursing students' experience with distance e-learning, nursing schools must try to reduce students' anxiety associated with COVID-19. |
| [3] | Social anxiety | SASE | Attempt to create a scale that measures social anxiety levels experienced during online learning. | Students | Among learners, the negative evaluation dimension measures their fears and feelings as they relate to trying to interact in an e-learning environment and being misjudged by someone else. |
| [8] | Computer anxiety | TRA and TAM | This study is designed to improve a usage intention model for e-learning systems. | Employees | Perceived ease of use and perceived usefulness of computers are affected by computer anxiety and self-efficacy. |
| [15] | Computer anxiety | TAM | The intentions of Saudi students to use an e-learning environment should be evaluated in terms of their enjoyment of the environment, their computer anxiety, their self-efficacy and their experience with the internet. | Students' universities' | Computer anxiety, self-efficacy and enjoyment significantly influenced the use of e-learning, whereas the internet experience failed to make a significant impact. Additionally, attitude was found to be a mediator of the relationship between perceived usefulness and perceived ease of use, as well as the behavioural intentions of the students. |

Prior studies also investigated the difference between social anxiety and computer anxiety. The e-learning environment seems to strongly impact social anxiety. The long periods of daily interactions may lessen the degree of social anxiety. Though the interactions may reduce social anxiety, it seems that they have no impact on computer anxiety. Studies

have agreed that learners' social anxiety is reduced due to repeated class interactions with teachers, group members and content [5]. This implies that a cooperative learning environment has a remarkable impact on anxieties because learners feel more relaxed and comfortable in collaborative information sharing [32]. The level of the presented material and the nature of taught courses can increase the level of social and computer anxiety. Poor language proficiency may lead to a high level of anxiety [33]. This is in line with other studies that emphasize learners' comprehension levels. Carrell [34] has proposed that advanced-level students were incapable of understanding the meaning of a reading text because of learners' lack of background knowledge.

The previous literature is filled with evidence illustrating the relationship between satisfaction and anxiety. It has been demonstrated that learners' level of anxiety is reduced whenever the level of satisfaction is high. Previous studies have also shown that the sudden shift to an e-learning environment has led to a higher level of anxiety. The correlation between anxiety and satisfaction was evident in such circumstances [35]. On the other hand, other studies have demonstrated that when e-learning is delivered purposefully and effectively, it leads to a positive effect on learners' satisfaction [36]. A similar result is shown by [37] who states that the satisfaction level of using e-learning and virtual classes is medium, showing variations in degrees from one learner to another [37].

## 3. Theoretical Framework

### 3.1. The Integration of Artificial Indigence Anxiety with Immersion, Interaction and Imagination

The spread of AI technology has led to a new environment that necessitates the need to put in mind the nature and sources of artificial intelligence anxiety. Recent studies have tackled this issue in a limited way, which does not reflect the widespread attention to using AI in various settings. Hence, the prior studies focused on the sources of anxiety and classify it on different levels. They have proposed various dimensions of AI depending on the theoretical model. Furthermore, the novelty of the AIA leads researchers to tackle its concept focusing on developing a standardized tool to measure this phenomenon by defining the construct of AIA [24,25]. During the past years, developers have started developing AI tools that can enhance the cognitive aspects of everyday classes. An example of AI that assists the cognitive aspect is the so-called expert model. It is an example of a "cognitive tutor" named SHERLOCK. The tutor presents tutorial actions associated with the concept of the "problem-solving" technique that is found in the space. It can transmit the problem to a kind of cognitive skill developer robot that can deal with students' problems, attempting to investigate which rules are being viewed as difficult. Recently, researchers are keener to adopt AI that can assist learners during the time in class by making a kind of global classroom that can be shared by students from various places to share their attitudes using innovative teaching styles. In addition, they attempt to solve the "at home" teaching problem that appears during the self-study technique. Virtual global conferencing is another example that shows how AI applications may be useful for teachers and the future development of smart content and personal development [38,39]. To be able to investigate learners' attitudes and intentions to use technology, the three features are included and integrated with AIA. Immersion is a subjective psychological response, not an objectively measurable property of a system. Another multi-dimensional construct, interaction, describes aspects of human-computer interaction as well as computer-mediated communication between humans. By virtue of the content of virtual environment applications, the imagination is stimulated by the capacity of the user's mind to perceive non-existent objects. It is the result of the combination of prior knowledge and recently acquainted knowledge [40,41]. The research model served as a guideline for formulating questionnaires and systematically performing statistical analyses to test the hypotheses. The three main features were examined to see whether they have a positive influence on the intention to use technology.

*3.2. Integration of Social Anxiety with Motivation and Satisfaction*

Social anxiety is identified as a feeling of worry or fear from doing something wrong and being judged negatively by others or giving a bad first impression. (American Psychiatric Association, 2000). It is the individual's fear of being watched all the time and assessed (criticized) negatively by other people. The individual with social anxiety is occupied by the fact that he or she is being continually watched by others. Being afraid of doing something wrong may result in judging him/her negatively. Anxious individuals fear making errors, performing embarrassing acts, looking bad and are lacking in social skills. People with social anxiety disorders differ physiologically, cognitively and behaviourally from ordinary people [42]. Individuals who experience this type of anxiety may prevent themselves from interacting with others or being involving in groups.

Experts have indicated a relationship between social anxiety and preferred communication method (online, face to face). Additionally, in [8], a negative relationship was discovered between the level of anxiety among learners and web-based learning continuance intentions. According to [43], individuals with social anxiety will avoid performing in front of others to reduce the social risk, and they will prevent themselves from interacting with others and show behaviours that may lead to people judging them. Therefore, social anxiety plays a crucial role in how one interacts with others, including the methods used to engage with them and the duration of those interactions.

According to the sociocultural constructivist theory, learners are active learners and self-motivated in the online environment. The natural differences between traditional classrooms and e-learning environments are different. However, studies have shown that it is not necessarily the case. Learners may not be remarkably motivated at the beginning of their study in an online environment because they may be anxious. As a result, anxious students are intrinsically less motivated and need more attention [44]. Students can be motivated in two ways: intrinsically or extrinsically. Intrinsic motivation occurs when students strive to learn because they are interested in it, or they want to achieve their own goals. However, according to [45], students who are intrinsically motivated will often exert the least amount of effort to reap the greatest benefit. Yet, using technology in the learning process does not necessarily motivate students. According to [46], the two types of motivation complement each other. As a result, learning also requires some driving force and extrinsic motivation, which can be represented through parents' support and attaining incentives. It turns out that motivation helps drive all the other processes involved in learning. However, ref. [47] claimed that students would expect rewards from others in exchange for the behaviours they exhibit. When students were motivated by extrinsic rewards, they became motivated by intrinsic motivation. A good motivation internalization process would be for students to acquire some accomplishment motivation or to transform it into the need for self-development during the learning procedure.

Satisfaction has been dealt with by different researchers, examining its effectiveness and relationship to various educational settings. In this respect, satisfaction can be referred to as the student's awareness of the importance of learning experiences in the learning setting. Student satisfaction considers vital factors when evaluating the effectiveness of a course or program. In addition to learning effectiveness, access, faculty satisfaction and institutional cost-effectiveness in e-learning, satisfaction are the five pillars of quality in e-learning; nevertheless, it is a complicated structure that includes numerous variables [48].

According to [49], flexibility, computer skills and usefulness are all factors associated with students' satisfaction with online learning. Student satisfaction is affected by a variety of elements in the online environment, involving teacher behaviour, trustworthy technology and engagement [50,51]. In addition, task worth, self-efficacy and multimedia tutoring quality, as well as students' social abilities all play major roles in the learning process [52]. Moreover, Liaw [52] found that student satisfaction is largely influenced by self-efficacy. In the long run, student satisfaction can lead to better motivation, learning and performance.

### 3.3. The Integration of Computer Anxiety with Self-Efficacy

E-learning platforms are affected by computer anxiety (CAX), which is defined as the apprehension or fear that learners may have during the use of technology. In fact, it is a kind of tendency that is developed by a learner due to their uneasiness over their experience in using computers [9,53]. The negative effects of computer technology have been the focus of much research since the 1970s. There have been a variety of studies examining the affiliation between computer anxiety and various variables. Past research has examined the connection between computer experience and computer anxiety, whereas other studies have investigated the effect of gender, age and personality traits on computer anxiety [13]. This anxiety may be extended to the use of e-learning as a new educational tool where computer anxiety can be regarded as a key factor affecting different types of intentions to use technology [54,55]. Prior studies have shown that the role of computer anxiety on students' attention may be emotional It is also assumed computer anxiety is considered as having a direct influence on the students' perception towards using e-learning platforms. The high level of anxiety toward computers may render the use of e-learning platforms and ultimately affect their intention to use e-learning tools in educational contexts [56]. Computer anxiety is connected with the perceived ease of use of technology that is reflected by learners during the teaching and learning process. This implies that when learners evaluate technology as effortless, their level of anxiety is less and vice-versa [14].

Self-efficacy can be defined as the person's faith in being able to execute the behaviours which will produce the desired outcome [57]. It is an explanation of one's ability to judge the quality of his/her execution to accomplish the desired goal. Additionally, other researchers have provided further clarification, emphasizing that self-efficacy is the belief in one's ability to perform tasks, and it has three elements: magnitude, strength and generality. Additionally, self-efficacy has a relationship to the internet experience. Self-efficacy is investigated by previous studies by connecting it to TAM constructs of the perceived ease of use and perceived usefulness, which have a direct relationship to the intention to use technology [15,58]. In this respect, self-efficacy was measured as an external variable in e-learning studies [59,60].

Rezaei et al. [61] investigated the technological intention to measure learners' intentions in e-learning at the university level in agricultural science. They extended the model by incorporating internet experience, self-efficacy and computer anxiety along with TAM variables. In this study, it was revealed that learners' desire to use e-learning was influenced by such factors as perceived usefulness, perceived ease of use, internet experience and computer self-efficacy. In contrast, students' behaviour intention to use e-learning was negatively correlated with their computer phobia.

### 3.4. Perceived Ubiquity and Innovativeness

Previous literature has shown that the concept of ubiquity has a distinctive value due to its ability to measure the effectiveness of technology anytime and anywhere. It is a remarkable factor that enables flexibility by providing users with time convenience and greater accessibility through spatial flexibility. Prior studies have shown ubiquity to be a crucial factor in decision-making behaviour and one that therefore needs to be incorporated into strategy formulation [62].

Similarly, innovativeness is a crucial factor that governs the intention to use technology. Innovativeness (INNO) can be defined as the learners' willingness to attempt to use new technology, which has a significant impact on users' intention to use the technology of teaching. The concept of willingness is governed by other factors that are either related to the personal characteristics of the learners themselves or the nature of the accepted technology [63,64].

### 3.5. Hypotheses of the Study

Based on the theoretical account that is given above, a group of hypotheses has been proposed to reflect the role of artificial intelligence and social anxiety and computer anxiety

in the intention to use technology. To evaluate the effectiveness of three types of anxiety empirically, a set of hypotheses has been formulated that show the relationship between anxiety and other external factors, namely, immersion, interaction, imagination self-efficacy, motivation and satisfaction (Figure 1). Accordingly, the following hypotheses can be proposed:

**H1.** *Artificial intelligence anxiety has a negative impact on immersion, interaction and imagination.*

**H2.** *Computer anxiety has a negative impact on self-efficacy.*

**H3.** *Social anxiety has a negative impact on motivation and satisfaction.*

**H4.** *Immersion, interaction and imagination have a significant impact on perceived ubiquity.*

**H5.** *Immersion, interaction and imagination have a significant impact on innovativeness.*

**H6.** *Motivation and satisfaction have a significant impact on perceived ubiquity.*

**H7.** *Motivation and satisfaction have a significant impact on innovativeness.*

**H8.** *Self-efficacy has a significant impact on perceived ubiquity.*

**H9.** *Self-efficacy has a significant impact on innovativeness.*

**H10.** *Perceived ubiquity has a significant impact on the intention to use technology.*

**H11.** *Innovativeness has a significant impact on the intention to use technology.*

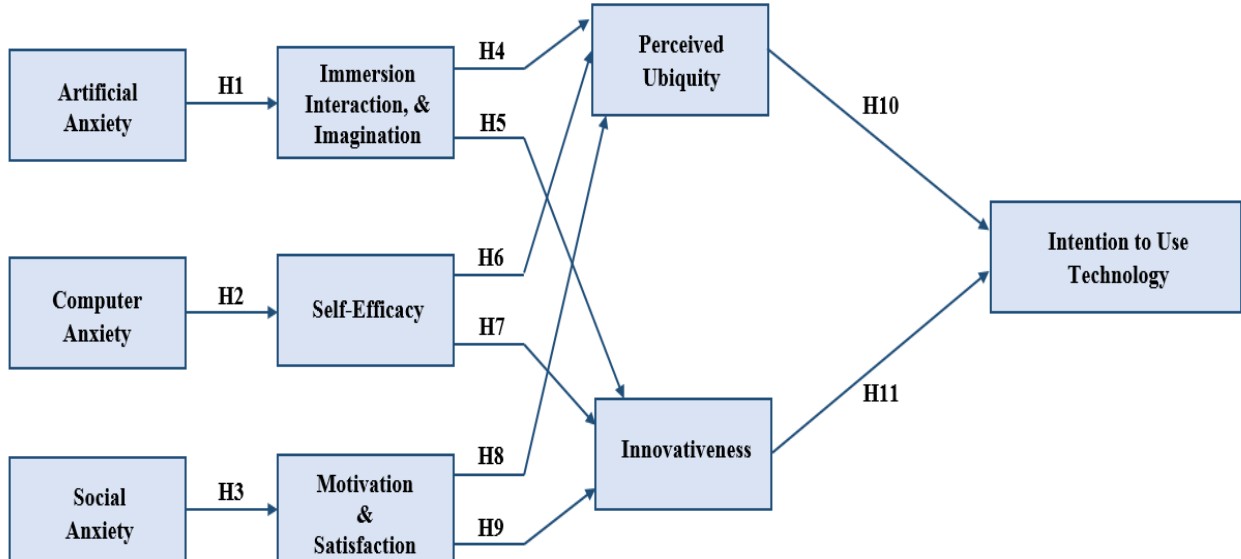

**Figure 1.** Research model.

## 4. Methodology

### 4.1. Data Collection

Data collection took place from 15 June to 20 February 2022 over the winter semester (2021–2022) at Al Buraimi University College in Oman using online surveys. A consent form and information sheet were included on the survey's opening page. Zero identifiable verification was needed to maintain the privacy of the data, and respondents were free to withdraw at any time without explanation. The survey's respondents received no remuneration of any kind for their participation. The University Students Research Evaluation Committee provided a letter of ethical permission for this work, enabling surveys to be undertaken inside Al Buraimi University College buildings. The group of researchers conducted a random distribution of 600 questionnaires. A total of 545 questionnaires have been answered by respondents, which represents a 91% response rate. Moreover, 55

questionnaires were rejected due to missing values. As a result, 545 questionnaires were usable. These accepted surveys had an appropriate sampling size, according to Krejcie and Morgan [65] (the expected sampling size for 306 respondents/1500 population). The difference is great between the sample size (545) and the minor requirements. Based on the previous fact, the sample size can be analyzed and evaluated using structural equation modelling [66], which was afterwards used to confirm the hypotheses. It is important to note that the previous theories (based on artificial intelligence and social and computer anxiety in educational contexts) were the foundation of our hypotheses. In order to evaluate the measurement model, the researchers used structural equation modelling (SEM) (SmartPLS Version 3.2.7). Advanced treatment was conducted with the help of the final path model.

*4.2. Students' Personal Information/Demographic Data*

Table 2 below illustrates the distribution of demographic/personal data which has been used for the purpose of the analysis. Based on the table, 47% out of the total number were males and the rest were females forming 53% of the total; adding to that, 60% of respondents were within the age range of 18–29 years old whereas the rest of the group was above 29. The total respondents comprised a group of students who had completed university degrees alongside those of an educated background. To put it differently, most of the students carry a Bachelor's degree, Master's degree or a doctoral degree, representing 67%, 22% and 11% of the group, respectively. Al-Emran and Salloum [67] suggested that if the respondents expressed their willingness for volunteering, there would be utilization of the "purposive sampling approach". As far as the sampling tool is concerned, the students included in the sample belonged to different universities, age groups and educational programs and levels. Furthermore, IBM SPSS Statistics ver. 23 was used for measuring the demographic data. Table 2 illustrates a deeper view of the respondents' demographic data.

**Table 2.** Demographic data of the respondents.

| Criteria | Factor | Frequency | Percentage |
|:---|:---|:---|:---|
| Gender | Female | 287 | 53% |
| | Male | 258 | 47% |
| Age | Between 18 to 29 | 329 | 60% |
| | Between 30 to 39 | 124 | 23% |
| | Between 40 to 49 | 86 | 16% |
| | Between 50 to 59 | 6 | 1% |
| Education qualification | Bachelor's | 365 | 67% |
| | Master's | 122 | 22% |
| | Doctorate | 58 | 11% |

*4.3. Study Instrument*

The survey instrument was adopted for the sake of validating the hypothesis. For the sake of precise measurement of the questionnaire's 9 constructs, the selected items consisted of 23 statements which were further added to the survey. The Table 3 illustrates the sources of these constructs, which helps in making the research more applicable. Furthermore, the researchers made amendments to the questions of prior research.

**Table 3.** Measurement Items.

| Constructs | Items | Definition | Instrument | Sources |
|---|---|---|---|---|
| Perceived Ubiquity | PUB1 | Perceived Ubiquity has a close relation with learners' attitude towards flexibility in time and space [68] (p. 98). It enhances the concept that the integration of various dimensions of time and space is possible flexibility [69]. | Using technology has no time and space limitation. | [68,69] |
| | PUB2 | | Using technology has a high level of flexibility which enables me to move freely. | |
| | PUB3 | | I am ready to use technology because its interrelated dimensions have no limit. | |
| Innovativeness | INNO1 | Innovativeness (INNO) refers to learners' willingness to attempt to use new technology, which has a significant impact on users intention to use the technology of teaching [63,64]. | Technology has innovative features that I like to use for my study. | [63,64] |
| | INNO2 | | Technology offers a unique, one-of-a-kind experience. | |
| | INNO3 | | I would like to use technology due to its innovative features. | |
| Artificial Intelligence Anxiety | AIA1 | AIA refers to the type of fear that learners may form after interacting with AI. Accordingly, it is a kind of multidimensional type of fear that operationally controls learners' perceptions [24]. | Artificial intelligence anxiety stops learners from using AI technology. | [24] |
| | AIA2 | | Artificial intelligence anxiety prevents learners from developing their skills in using AI technology. | |
| Immersion, Interaction and Imagination | III1 | These are three dimensions that can affect any artificial intelligence technology because it evaluates a dynamic virtual world which is associated with real-time interaction. Learners' immersion while using the technology and their imagination is based on their daily interaction and form the essence of the intention to use technology. The imaginary factor affect is perceptual knowledge, which allows for constructivist learning [40,70]. | Technology helps learners to be live in daily learning classes by reducing their artificial intelligence anxiety. | [40,70] |
| | III2 | | Technology permits learners to interact freely without time or space limitations so it reduces learners' artificial intelligence anxiety. | |
| | III3 | | Technology allows learners to use their imagination freely, which helps in minimizing artificial intelligence anxiety. | |
| Social Anxiety | SA1 | It is the individual's fear of being watched all the time and assessed (criticized) negatively by other people. The individual with social anxiety is occupied by the fact that the he or she is being continually watched by others. Being afraid of doing something wrong may result in judging him/her negatively [42]. | Social anxiety prevents learners from communicating with others via technology. | [42] |
| | SA2 | | Social anxiety reduces my participations when I am using technology. | |
| Motivation and Satisfaction | MS1 | Motivation can be intrinsically or extrinsically oriented and students present motivations along a continuum ranging from lack of control to self-determination: from no motivation at all (motivation), to externally oriented motivation (extrinsic) to internally oriented motivation (intrinsic). Satisfaction has a close relationship with intrinsic motivation that leads to pleasure and satisfaction obtained from learners' participation | Learners are able to communicate with less anxiety if they feel motivated and satisfied. | [71] |
| | MS2 | | Learners are able to participate with less anxiety if they feel they are satisfied and motivated. | |
| | MS3 | | Motivation and satisfaction can reduce learners' anxiety in using technology. | |
| Computer Anxiety | CA1 | Computer anxiety refers to instances when learners develop a special kind of fear and apprehension that prevents them from using or developing computer-based skills [24]. | Computer anxiety prevents learners from developing their technology skills. | [24] |
| | CA2 | | Computer anxiety is an obstacle in the way of using new technology. | |
| Self-efficacy | SEFC1 | Self-efficacy refers to one's judgment of one's ability to complete tasks. Users may reflect if they can control the technology with minimal effort. Learners have a foundational judgment about their ability to use technology. | It is easy for learners to complete their tasks if they have a lesser level of computer anxiety. | [72] |
| | SEFC2 | | Learners finalize their assignments if they have good computer skills. | |
| | SEFC3 | | Learners complete their daily homework if they feel comfortable with using computer skills. | |

**Table 3.** *Cont.*

| Constructs | Items | Definition | Instrument | Sources |
|---|---|---|---|---|
| Intention to Use Technology | IUT1 | Intention to use is used as a variable that shows users' willingness to accept the technology. The theory of intention to use is developed on social psychological behavior that shows users' willingness to perform an action or adopt a behaviour [73,74]. | I intend to use technology in the future because it is highly flexible. | [73] |
| | IUT2 | | I expect that I will continue to use the technology because it has innovative features. | |

*4.4. Pilot Study of the Questionnaire*

The questionnaire items were tested for reliability in a pilot study. In this pilot study, 60 students were randomly selected from the determined population. Taking 10% of the total sample size into consideration, 600 students were selected as the sample size, and the research guidelines were highly emphasized using Cronbach's alpha test via IBM SPSS Statistics version 23, and the pilot study results were evaluated for internal reliability, which led to acceptable conclusions. Based on the stated trend of social science research, a reliability coefficient of 0.70 is considered acceptable [75]. Table 4 presents the Cronbach's alpha values in terms of the subsequent 5 measurement scales.

**Table 4.** Cronbach's Alpha values for the pilot study (Cronbach's Alpha ≥ 0.70).

| Construct | Cronbach's Alpha |
|---|---|
| AIA | 0.803 |
| CA | 0.822 |
| III | 0.799 |
| IUT | 0.892 |
| INNO | 0.793 |
| MS | 0.872 |
| PUB | 0.872 |
| SA | 0.815 |
| SEFC | 0.821 |

*4.5. Survey Structure*

The survey that was distributed includes three parts:

- The first part is concerned with the respondents' personal data.
- The second part has two items that are related to the general question related to "Intention to Use Technology".
- The third part embraces 21 items that have detailed statements about "Perceived Ubiquity, Innovativeness, Artificial Intelligence Anxiety, Immersion, Interaction and Imagination, Social Anxiety, Motivation and Satisfaction, Computer Anxiety and Self-efficacy".

For measuring the 23 items, a five-point Likert Scale will be considered with the following options: strongly disagree (1), disagree (2), neutral (3), agree (4) and strongly agreed (5).

## 5. Findings and Discussion

*5.1. Data Analysis*

The current study has been developed depending on the use of the partial least squares-structural equation modelling (PLS-SEM) through SmartPLS V 3.2.7 [76]. The data was collected and analyzed using a two-step assessment approach that incorporates the measurement model and the structural model [77]. The PLS-SEM was chosen in this scientific research paper for several reasons.

Firstly, the total aim of the study is work on a current theory, so the priority is given to the preference that suits the PLS-SEM as an analysis tool [78]. The second step is to use the PLS-SEM to sufficiently handle the exploratory research along with its complex models [79]. The third step is to use the PLS-SEM to signify the analysis of the entire model as one unit rather than making subdivisions out of it [80]. Finally, PLS-SEM has the power to provide the analysis with structural and measurement models, because of its accurate measurements [81].

### 5.2. Convergent Validity

For assessing the measurement model, [77] suggested construct reliability (which includes Cronbach's alpha (CA), Dijkstra-Henseler's (PA) and composite reliability (CR)) and validity (which includes discriminant and convergent validity). For determining the construct reliability, Cronbach's alpha (CA) was found to be within the range of 0.778–0.899, with respect to Table 4. The threshold value (0.7) is lower than these figures [82]. According to Table 4, the results show that the composite reliability (CR) values range from 0.800 to 0.892, which exceed the threshold value [83]. Instead, researchers should use the Dijkstra-Henseler's rho (ρA) reliability coefficient for evaluating and reporting construct reliability [84]. As with CA and CR, the reliability coefficient ρA should be at least 0.70 (exploratory research) and 0.80 or 0.90 (advanced research stages) [82,85,86]. Table 4 also shows that 0.70 is the minimum reliability coefficient ρA of all measurement constructs. These results confirm the construct reliability, and each construct was considered to be free from errors, ultimately.

When it comes to the measurement of convergent validity, it is necessary to test the mean variance extracted (AVE) and factor loading [77]. Apart from that, Table 4 suggests that each factor loading value exceeded the threshold value of 0.7. Other than that, according to the Table 1 results, the AVE values ranged from 0.528–0.783, which are expected to exceed the '0.5' threshold value. On the basis of these following results, it is possible to achieve the convergent validity.

### 5.3. Discriminant Validity

To measure discriminant validity, it was suggested to consider two criteria that include the Heterotrait-Monotrait ratio (HTMT) and Fornell–Larker criterion [77]. Table 5 findings suggest that the Fornell–Larker condition confirms the requirements because each AVE and their square roots exceed its correlation with other constructs [87].

**Table 5.** Convergent validity results.

| Constructs | Items | Factor Loading | Cronbach's Alpha | CR | PA | AVE |
|---|---|---|---|---|---|---|
| Artificial Intelligence Anxiety | AIA1 | 0.842 | 0.857 | 0.817 | 0.880 | 0.683 |
| | AIA1 | 0.818 | | | | |
| Computer Anxiety | CA1 | 0.888 | 0.841 | 0.886 | 0.869 | 0.627 |
| | CA2 | 0.898 | | | | |
| Immersion, Interaction and Imagination | III1 | 0.895 | 0.821 | 0.834 | 0.828 | 0.669 |
| | III2 | 0.726 | | | | |
| | III3 | 0.841 | | | | |
| Intention to Use Technology | IUT1 | 0.759 | 0.839 | 0.831 | 0.829 | 0.670 |
| | IUT2 | 0.881 | | | | |
| Innovativeness | INNO1 | 0.855 | 0.842 | 0.823 | 0.825 | 0.650 |
| | INNO2 | 0.930 | | | | |
| | INNO3 | 0.918 | | | | |

**Table 5.** *Cont.*

| Constructs | Items | Factor Loading | Cronbach's Alpha | CR | PA | AVE |
|---|---|---|---|---|---|---|
| Motivation and Satisfaction | MS1 | 0.917 | 0.899 | 0.892 | 0.882 | 0.783 |
| | MS2 | 0.802 | | | | |
| | MS3 | 0.761 | | | | |
| Perceived Ubiquity | PUB1 | 0.868 | 0.778 | 0.800 | 0.834 | 0.528 |
| | PUB2 | 0.836 | | | | |
| | PUB3 | 0.702 | | | | |
| Social Anxiety | SA1 | 0.842 | 0.859 | 0.853 | 0.855 | 0.626 |
| | SA2 | 0.873 | | | | |
| | SA3 | 0.817 | | | | |
| Self-efficacy | SEFC1 | 0.851 | 0.810 | 0.812 | 0.822 | 0.731 |
| | SEFC2 | 0.750 | | | | |
| | SEFC3 | 0.761 | | | | |

Tables 6 and 7 show the HTMT ratio findings, which illustrates that the value of each construct is lower than the '0.85' threshold value [88]. As a result, there is the presence of the HTMT ratio. As a result of these findings, discriminant validity can be calculated. Based on the analysis results, there were no reliability or validity issues with the measurement model. Because of it, the collected data can be further used for evaluating the structural model.

**Table 6.** Fornell-Larker Scale.

| | AIA | CA | III | IUT | INNO | MS | PUB | SA | SEFC |
|---|---|---|---|---|---|---|---|---|---|
| AIA | **0.846** | | | | | | | | |
| CA | 0.469 | **0.860** | | | | | | | |
| III | 0.396 | 0.267 | **0.819** | | | | | | |
| IUT | 0.555 | 0.351 | 0.250 | **0.859** | | | | | |
| INNO | 0.551 | 0.405 | 0.406 | 0.330 | **0.822** | | | | |
| MS | 0.489 | 0.360 | 0.388 | 0.218 | 0.519 | **0.889** | | | |
| PUB | 0.283 | 0.111 | 0.248 | 0.617 | 0.215 | 0.283 | **0.811** | | |
| SA | 0.325 | 0.246 | 0.209 | 0.508 | 0.299 | 0.325 | 0.246 | **0.809** | |
| SEFC | 0.350 | 0.222 | 0.280 | 0.761 | 0.296 | 0.285 | 0.350 | 0.222 | **0.830** |

**Table 7.** Heterotrait-Monotrait Ratio (HTMT).

| | AIA | CA | III | INNO | IUT | MS | PUB | SA | SEFC |
|---|---|---|---|---|---|---|---|---|---|
| AIA | | | | | | | | | |
| CA | 0.442 | | | | | | | | |
| III | 0.413 | 0.350 | | | | | | | |
| IUT | 0.478 | 0.415 | 0.657 | | | | | | |
| INNO | 0.520 | 0.434 | 0.612 | 0.659 | | | | | |
| MS | 0.471 | 0.559 | 0.582 | 0.564 | 0.597 | | | | |
| PUB | 0.482 | 0.502 | 0.631 | 0.603 | 0.664 | 0.583 | | | |
| SA | 0.236 | 0.079 | 0.267 | 0.163 | 0.276 | 0.292 | 0.160 | | |
| SEFC | 0.173 | 0.339 | 0.261 | 0.250 | 0.362 | 0.372 | 0.325 | 0.451 | |

*5.4. Hypotheses Testing Using PLS-SEM*

For determining whether the structural model's theoretical constructs are interdependent, there was utilization of the structural equation model alongside Smart PLS with maximum likelihood estimation [89–91]. Based on the analysis, the proposed hypotheses were supported and completed. As shown in Figure 2 and Table 8, both of them illustrate the high predictive power of the model [92], i.e., there was 80.7% variance within Intention to Use Technology (IUT).

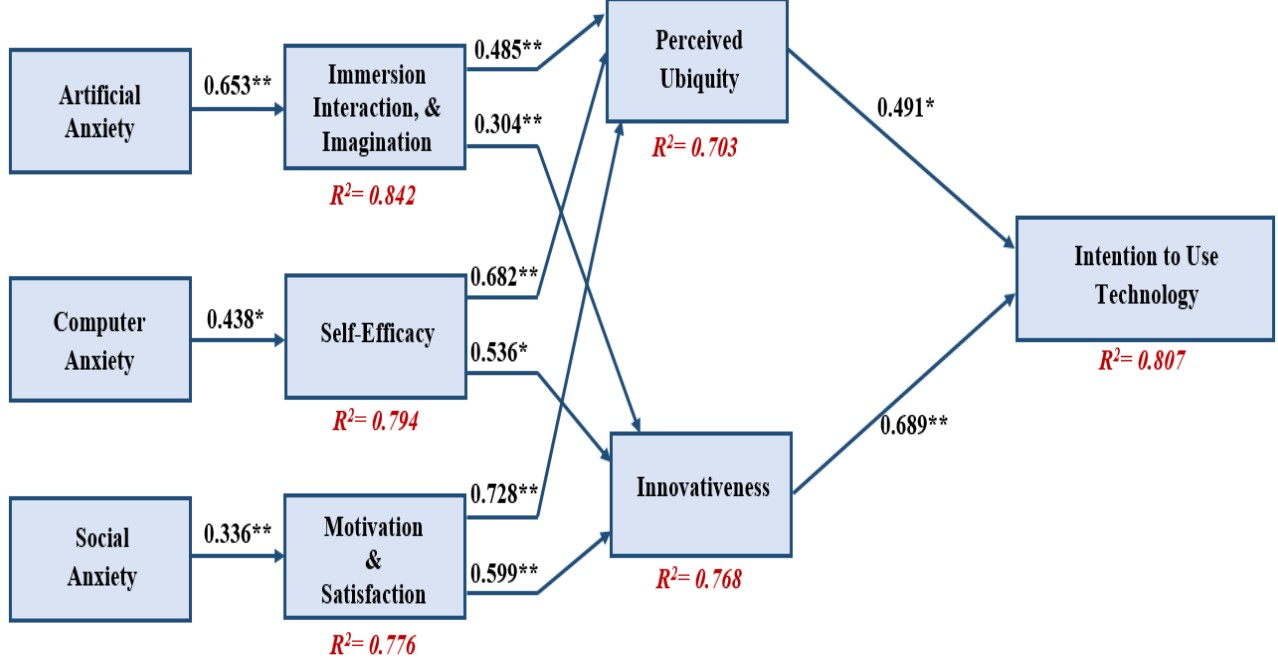

**Figure 2.** Path coefficient of the model (significant at $p$ ** $< = 0.01$, $p$ * $< 0.05$).

**Table 8.** $R^2$ of the endogenous latent variables.

| Construct | $R^2$ | Results |
|---|---|---|
| III | 0.842 | High |
| INNO | 0.768 | High |
| IUT | 0.807 | High |
| MS | 0.776 | High |
| PUB | 0.703 | High |
| SEFC | 0.794 | High |

In Table 9, the beta (β) values, *t*-values and *p*-values for all of the proposed hypotheses have been dealt with depending on the statistical findings, with the help of the PLS-SEM technique. It is quite clear that the hypotheses that have been stated previously are supported and confirmed, which enhances the empirical data and the suggested hypotheses.

Table 8 illustrates the beta (β) values, *t*-values and *p*-values for each of the proposed hypotheses based on the generated results through the PLS-SEM technique. Based on the results, it is obvious that all the hypotheses are confirmed. Based on that, the data analysis hypotheses H1, H2, H3, H4, H5, H6, H7, H8, H9, H10 and H11 were supported by the empirical data. The results showed that "Immersion, Interaction and Imagination (III)" significantly influenced artificial anxiety (AI) (β = 0.653, $p < 0.01$), supporting hypothesis H1. Computer Anxiety (CA) was determined to be significant in affecting Self–Efficacy (SEFC) (β = 0.438, $p < 0.05$), supporting hypothesis H2. The results also showed that

Social Anxiety (SA) significantly influenced "Motivation and Satisfaction (MS)" ($\beta = 0.336$, $p < 0.01$), supporting hypothesis H3. "Immersion, Interaction and Imagination (III)" has significant effects on Perceived Ubiquity (PUB) ($\beta = 0.485$, $p < 0.001$) and Innovativeness (INNO) ($\beta = 0.304$, $p < 0.001$), respectively; hence, H4 and H5 are supported. Self-efficacy (SEFC) and Motivation and Satisfaction (MS) had significant effects on Perceived Ubiquity (PUB), ($\beta = 0.682$, $p < 0.001$) and ($\beta = 0.728$, $p < 0.001$), respectively; hence, H6 and H8 are supported. The findings also revealed that Self-efficacy (SEFC) and Motivation and Satisfaction (MS) had significant effects on Innovativeness (INNO), ($\beta = 0.536$, $p < 0.05$) and ($\beta = 0.599$, $p < 0.001$), respectively; hence H7 and H9 are supported. Finally, the relationship between Perceived Ubiquity (PUB) and Innovativeness (INNO) had significant effects on Intention to Use Technology (IUT) ($\beta = 0.491$, $p < 0.05$) and ($\beta = 0.689$, $p < 0.001$), respectively; hence, H11 and H12 are supported.

**Table 9.** Hypotheses testing of the research model (significant at $p$ ** $< = 0.01$, $p$ * $< 0.05$).

| H | Relationship | Path | *t*-Value | *p*-Value | Direction | Decision |
|---|---|---|---|---|---|---|
| H1 | AIA → III | 0.653 | 8.431 | 0.004 | Positive | Supported ** |
| H2 | CA → SEFC | 0.438 | 5.542 | 0.022 | Positive | Supported * |
| H3 | SA → MS | 0.336 | 7.153 | 0.003 | Positive | Supported ** |
| H4 | III → PUB | 0.485 | 16.508 | 0.000 | Positive | Supported ** |
| H5 | III → INNO | 0.304 | 14.688 | 0.000 | Positive | Supported ** |
| H6 | SEFC → PUB | 0.682 | 6.953 | 0.006 | Positive | Supported ** |
| H7 | SEFC → INNO | 0.536 | 4.336 | 0.041 | Positive | Supported * |
| H8 | MS → PUB | 0.728 | 6.883 | 0.003 | Positive | Supported ** |
| H9 | MS → INNO | 0.599 | 16.515 | 0.000 | Positive | Supported ** |
| H10 | PUB → IUT | 0.491 | 4.350 | 0.034 | Positive | Supported * |
| H11 | INNO → IUT | 0.689 | 13.366 | 0.000 | Positive | Supported ** |

## 6. Discussion of Results

The main contribution of the current study is that it focuses on the integration of AIA with other types of anxiety such as complexity and social anxiety. Thus, the findings help in creating a relationship between AIA and other motivational external factors. The development of the AIA conceptual model represents a significant step in the theoretical development process related to AIA and AI adoption. Accordingly, the paper adds a new dimension to its conceptual model by integrating crucial and decisive factors that can increase or decrease the level of anxiety, and it paves the way for empirical validation of the adoption of AI in the educational sector.

Advances in artificial intelligence (AI) have sparked the development of educational artificial intelligence tools. Teachers can use AI to formulate better pedagogical decisions for their students. AI tools are hardly integrated into teaching, and little is known about their perceptions. The results indicate that educators, teachers and students are more likely to integrate AI in their educational settings. AI dwells upon the fact that it is an innovative teaching tool in addition to its close correlation with perceived ubiquity and innovativeness. These two factors assist the use of AI tools and are crucial determinants to be considered when explaining teachers' and students' acceptance of AI. The previous studies have addressed the users' AI concerns deeply, trying to find the best solutions to overcome these difficulties.

The results of the literature have shown that several external factors may affect the acceptance of AI, particularly perceived ease of use and perceived usefulness. However, some other studies reached certain conclusions that are not in agreement with the generalizations of the current study. It revealed disconfirmation effects regarding the acceptance

of the AI system, namely, the differences between expectations around technology prior to use and after usage [93].

An immediate and direct correlation is found between AL and the factors of immersion, interaction and imagination. The current results are supported by previous studies that adhere to the fact that immersion can be positioned as an external exogenous variable that impacts interaction and imagination in the high-immersion virtual reality technology acceptance model based on these results [94–96]. These findings provide theoretical support for predicting users' acceptance of IA technology in educational settings. Other studies have shown that there are reasons behind the effectiveness of immersion factor in technology acceptance. A study by [97] has proposed that immersive environments can allow educators and students to fully experience a real or artificial environment, increasing isolated sensory booths' external validity. Hence, the quality of knowledge gained using AI as compared to others will be different as far as the immersion aspect is concerned. It differs in terms of natural environment, benefits and level of engagement. Based on that, AI's immersive and interactive features reproduce the physical properties of external reality, and the synthetic presence is derived from mediated perceptions of those characteristics in simulated environments.

Among all the proposed independent factors, the most influential determinant of predicting the users' acceptance was found to be self-efficacy, motivation, satisfaction and how easily the AI tools are constructed. Self-efficacy, motivation and satisfaction are remarkable factors that enhance users' usage of technology. The previous literature has supported the results of the current paper stating that learning motivation, self-efficacy and satisfaction significantly affect the integration of technology. It suggests that the educational setting has a close relationship with computer self-efficacy and satisfaction. Whenever the level of motivation and satisfaction is high, students' willingness to continue using technology will be more different and useful. Studies have made the correlation with these variables and TAM constructs showing that the correlation between these factors and TAM is crucial to understanding the type of determinates that encourage users to use technology [98–100]. However, this study deviates from previous literature in the conceptual model and the results. The study dwells on the psychological aspects of the users by emphasizing the relationship between self-efficacy and computer anxiety. The study has shown that students' computer anxiety significantly affects their self-efficacy. The higher the level of computer anxiety, the more difficult is faced to reduce self-efficacy. Similarly, a relationship is created between social anxiety and motivation and satisfaction. It suggests that in an educational setting, there is a remarkable impact of social anxiety on satisfaction and motivation and a highly significant effect on the integration of AI as an educational tool. Previous studies have revealed that social anxiety has a close relationship with perceived users' willingness to accept and continue using AI. Social anxiety is in turn affected by other factors such as the hedonic value of AI. Some studies have argued that understanding the importance of social anxiety is crucial to understanding users' intentional behaviour. Similarly, AI assistant advantages are important factors affecting the utilitarian/hedonic value perceived by users, which further influences user willingness to accept AI assistants. The relationships between AI assistant advantages and utilitarian and hedonic values are affected differently by social anxiety. Marketers and managers in the AI context can refer to study methods to help improve AI assistants and develop more effective marketing strategies for product promotion [101–103].

### 6.1. Theoretical and Practical Implications

The current paper implies that the finding will facilitate the work of both AI developers and practitioners that show interest in applying and implementing AI in the educational sector. On the other hand, scholars and educators will receive a benefit from the results that focus on the importance not only of AIA but the correlation between AIA and computer anxiety and social anxiety. Hence, scholars and educators are encouraged to employ AIAS in AI and learning environments in their future learning styles.

The findings of this study provide a preliminary insight into the relationship between AIA and theories in the field of information technology. The importance lies in establishing a close relationship between educational concepts such as motivation and satisfaction with other technology-based information such as self-efficacy and innovativeness. The AIAS demonstrates satisfactory reliability and validity across various AI technologies/products.

The theoretical implication lies in choosing the appropriate AI that suits certain theory-based courses. It is an essential recommendation to specify the studies on education that will fit within AI. The new programs should consider what will be merged, bearing in mind the differences among the different majors and the nature of offered courses, and distorting some educational and cultural aspects in any future AI tools that will be implemented in educational settings.

The results provide some practical implications for the acceptance of AI at the university level. Firstly, IA technology should only be adopted after a careful assessment of learning domains and the tasks that have been conducted. Generally, AI can be used in subjects such as chemistry, geology, astronomy, surgery, history, culture and safety education to teach abstract concepts, procedural knowledge, attitude and authentic problem-solving. Furthermore, student engagement with AI technology can only be seen as part of the learning process, since other key learning activities might be needed outside of the IA technology environment. For example, individuals or groups can make use of AI in practical courses that enhance other students' practice and engagement, hence supporting the final performing outcomes. Third, the all-purpose use of AI is under consideration and can be described as desirable. For instance, direct instruction can be the sole pedagogical purpose of implementing convenient AI tools which can better fulfil educational needs and goals by providing supplementary activities. The importance of assessment in AI-based instruction cannot be overstated because it is often neglected as an integral component of the learning process.

### 6.2. Managerial Implications

As a result of this study, developers and creators of artificial intelligence will know which technologies to address in future development of AI in innovative educational settings. AI tools and the way AI and humans collaborate can be improved through these findings, which provide insightful suggestions for educators, developers and teachers. AI is generally considered cutting-edge technology, but certain considerations must be taken into account. First, consideration should be given to users' learning experiences. AI has a close relationship with social and computer anxiety, which may affect the level of performance. The lack of these considerations may affect the learning assessment, collaboration, educational equipment and the level of performance, limiting the possible learning activities within the AI environment. Furthermore, AI users face persistent challenges due to the social anxiety obstacle which negatively affects the physical discomfort and safety of the environment. These consequences may hinder the adoption of AI in a more regular curriculum instruction-based environment. Consequently, we suggest that AI features should be redefined to suit learning goals and assessment methods. It may be possible to use AI tools in educational settings as part of an advanced and continuous process when AI tools have the characteristics of full immersion, multisensory interaction and imagination.

### 6.3. Limitations of the Study and Future Studies

The study revealed various limitations and the need for future developments. First, the current study has developed a survey that investigates the relationship between artificial intelligence and other crucial factors. The limitation lies in the type of the proposed independent variable, because most of them are related to different types of anxiety. Therefore, future research may add other aspects to the questionnaire that addresses new opportunities and challenges in the questionnaire parts. Surveys should therefore be developed that address as much as possible the aspects that highlighted innovative chances and expected risks by specifying new and never explored fields of acceptance. Second, the study dwelled

on views from educators, teachers and students from a specific county in the Arab world. Their backgrounds and views may differ in education and students from different cultures and backgrounds. Hence, it will be essential to carry out studies on education in other parts of the world, considering what has emerged [104,105], also bearing in mind the differences in technology readiness and satisfaction [106–108]. AI applications concerning psychological factors in non-educational settings, especially in the banking field and health domains should be considered. Efforts should be made to manage customers' as well as doctors' relationships with AI applications. The factor of psychological safety has to be integrated carefully into AI applications. Measures can be implemented to foster a sense of psychological safety which can be implemented by improving the responsiveness of AI services through service process optimization of AI images for customers to view in terms of visual esthetics, anthropomorphic sounds and intimate contact.

## 7. Conclusions

In recent years, we have seen a reduction in the use of traditional methods of learning (traditional classroom) as a result of the increasing use of technology in learning (e-learning). E-Learning has caused more and more learners to experience anxiety, including social anxiety and computer anxiety. As a result of this increased anxiety, learners' results have dramatically declined, as well as their engagement levels in the classroom. Computer anxiety and social anxiety play an important role in the e-learning environment. These two factors play an important role in learners learning outcomes. As a result, it was shown that cooperative learning environments have a remarkable impact on anxiety because learners feel more relaxed and comfortable when sharing information collaboratively; however, previous studies have shown that traditional classrooms may increase the level of anxiety in comparison with e-learning environments. The interpersonal interactions of learners in e-learning environments can positively impact learners' level of comprehension, leading to a successful learning environment. The levels of social anxiety in learner–learner interaction and learner–instructor interaction may be reduced when the level of interaction is higher. Additionally, computer anxiety is related to individual anxiety about using technology, and it does not have any interaction with social anxiety. This study yields a conceptual model to identify the relationship between computer anxiety and social anxiety with other external factors, including self-efficacy, motivation and satisfaction in an e-learning environment. As the theoretical framework of the current study shows, a general attitude was predicted in the relationship between social anxiety and computer anxiety with other external factors such as self-efficacy, motivation and satisfaction. The study hypothesized seven different hypotheses for each relation. As previously stated, the questionnaire will be distributed among a group of graduate students to identify the levels of social anxiety and computer anxiety in learner–learner interaction and learner–instructor interaction.

**Author Contributions:** Conceptualization, M.A.A. and S.A.S.; methodology, S.T. and R.A.; software, F.H.; validation, A.L. and M.A.; formal analysis, F.A.E.-Q., A.A.M. and T.A. investigation, A.B.A. and R.S.A.-M.; resources, A.L.; writing—original draft preparation, M.A.A. writing—review and editing, S.A.S.; visualization, M.A.A.; supervision, M.A.A.; project administration, M.A.A.; funding acquisition, M.A.A. All authors have read and agreed to the published version of the manuscript.

**Funding:** This work was funded by King Faisal University (Project No. GRANT 875) and Princess Nourah bint Abdulrahman University (PNURSP2022R236).

**Data Availability Statement:** Not Applicable.

**Acknowledgments:** This work was supported through the Annual Funding track by the Deanship of Scientific Research, Vice Presidency for Graduate Studies and Scientific Research, King Faisal University, Saudi Arabia (Project No. GRANT 875); and Princess Nourah bint Abdulrahman University Researchers Supporting Project Number (PNURSP2022R236), Princess Nourah bint Abdulrahman University, Riyadh, Saudi Arabia.

**Conflicts of Interest:** All authors declare no conflict of interest.

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
