# Peer review of "Examining the Impact of Artificial Intelligence and Social and Computer Anxiety in E-Learning Settings: Students’ Perceptions at the University Level"

_electronics, doi:10.3390/electronics11223662_

Round 1

Reviewer 1 Report

The objective of this research is to examine the perceptions of the learners towards these types of anxiety. 

Authors claim that “The learners, who use any tool in an e-learning setting for the first time, are confused and frustrated”. However, in 10 years the MIT Open learning platform had 1 million students, apparently without major problems. Udacity has its whole business model focused on on-line learning and has reported an increasing profit curve. Maybe there are lessons to be learned from these two examples that would help the authors to improve their research? 

Figure 1: Research Model is interesting. However, it lacks a solid scientific base in order to gain more relevance. It is based on students’ perceptions, which to the best of my knowledge, is too weak to be presented as a solid model.

Moreover, when AI is involved, the Ethical concerns must be taken into account. However, there is not a single word in the text, and it is a CRUCIAL part of any AI-related research work.

In the “Theoretical and practical implications”, the authors recommend that “Firstly, IA technology should only be adopted after a careful assessment of 569 learning domains and tasks has been conducted” but this is a kind of “common sense knowledge” about TI tools in general and, more specifically, about AI tools since they can be strongly biased.

Section 6.3 presents a “cultural limitation”. Therefore, this difference shall be part of the AI tools  development, meaning that AI tools must be culturally tailored or they will have no use.

Still in this section authors argue that “Their background and views may differ in education and students from different cultures and backgrounds” and my point is: What if this limitation put in danger the results presented here? 

In section 6.2 authors state that “As a result of this study, developers and creators of artificial intelligence will know which technologies to address in future development of AI in innovative educational settings”. Statement is nice but not enough. The development of AI tools require much more than the results of a questionnaire-based research. MIT technology Review published a paper stating that “The EU wants to put companies on the hook for harmful AI - A new bill will allow consumers to sue companies for damages—if they can prove that a company’s AI harmed” them, just to illustrate the point I raised.

Table 1: Type of Technology - mismatch with the content presented (e.g. “Analyzing related literature”)

Minor errors:

Different paragraph formats along the text. 

Check the writing, there are minor errors in the paragraphs (e.g. use of “.”, and alike).

Author Response

The authors are really very grateful to the valued feedback and comments raised by the reviewer which really assist them to significantly enhance this work and its presentation. The productive and valuable remarks enable us to update many parts of the paper as shown by the responses to each comment. Our responses are mentioned below under each comment raised by the reviewer and it is written in (Times New Roman, red color). Besides, all the updated parts in the manuscript were highlighted in yellow color in order to be easily tracked by the reviewers.

Reviewer 2 Report

In line 175 the term "Artificial Indigence Anxiety" seems to be incorrect. The text has inumerous mistakes and must be fully and carefully revised.

Authors state in abstract that in this study they have proposed a theoretical framework to investigate the role of social anxiety and computer anxiety in e-learning environments in the Gulf region. They have examined how different variables such as satisfaction, motivation and self-efficacy can affect negatively or positively these two types of anxiety.

In introduction they introduce a new term Artificial intelligence anxiety. But, its not clear in the paper how AI introduces anxiety in learning environments, why it happens, and which techniques or AI applications are nocive. In Table 1, authors state that "the increasing importance of artificial intelligence necessitates  the need to reduce the anxiety  that appears as a result of using AI technologies". But, its not possible to find out in the paper why AI causes the appearence of anxiety in students. 

In addition, it is stated that "AIA refers to the type of fear that learners may form after interacting with AI. Accordingly, it’s a kind of multidimensional type of fear that operationally controls learners’ perceptions" But, it is not depicted which application of AI causes fear, and how to mitigate it. 

In addition authors affirm that "The theoretical implication lies in choosing the appropriate AI that suits certain theory based courses." But, the paper does not bring this relation between AI techniques and courses. The paper doesn't even cite the name of any AI technique. 

Therefore, it is really difficult to find out the real contribuition of the paper and how it can provide some advance in the state-of-the-art in this research field.

Author Response

(The authors gave the same response as above.)

Reviewer 3 Report

In general, the paper has a good flow and it is easy to follow. 

My comments are minor.

In line 89, please introduce the term AIA as it is never used before.

In line 122, also introduce the term TAM.

In line 124, introduce the term TRA.

In line 136, what is SASE?

The Procedure section which is missing. I would expect from authors to provide a thorough description of the study's procedure.

Finally, the authors might want to rewrite their recommendations for future studies as in its current version it is not adequate for researchers who are interested in these psychological factors.

Author Response

(The authors gave the same response as above.)

Round 2

Reviewer 1 Report

The authors addressed most of my comments, I must say, and I really appreciate that.

However, there is still one point that authors shall think about more deeply: the fact that DE is not "as good as" offline education (it was written in the answers provided to my comments). There are already good examples around the world showing that, based on the right methodologies and using the appropriate techniques, the online environment can be as good as the offline (e.g. MIT open learning platform, that celebrated 10 million students in 10 years of operation - impossible to be achieved by any offline institution in the world).

Author Response

The authors are really very grateful to the feedback and comments raised by the reviewer which really assist them to significantly enhance this work and its presentation. The productive and valuable remarks enable us to update many parts of the paper as shown by the responses to each comment. Our responses are mentioned below under each comment raised by the reviewer and it is written in (Times New Roman, red color). Besides, all the updated parts in the manuscript were highlighted in yellow color in order to be easily tracked by the reviewers.

Reviewer 2 Report

The paper is interesting, but for me, more practical analysis and implications must be explored:

It is stated that "AIA refers to the type of fear that learners may form after interacting with AI. Accordingly, it’s a kind of multidimensional type of fear that operationally controls learners’ perceptions" But, it is not depicted which application of AI causes fear, and how to mitigate it. 

In addition authors affirm that "The theoretical implication lies in choosing the appropriate AI that suits certain theory based courses." But, the paper does not bring this relation between AI techniques and courses. The paper doesn't even cite the name of any AI technique. 

Therefore, it is really difficult to find out the real contribuition of the paper and how it can provide some advance in the state-of-the-art in this research field.

For me, its a very interesting study that is in beginning, and must be deepen in order to bring more solid contributions.

Author Response

(The authors gave the same response as above.)

Reviewer 3 Report

I am pleased with the revised version. Please fix the text formating.

Author Response

(The authors gave the same response as above.)
